# The hallmarks of a tradeoff in transcriptomes that balances stress and growth functions

Christopher Dalldorf,[1] Kevin Rychel,[1] Richard Szubin,[1] Ying Hefner,[1] Arjun Patel,[1] Daniel C. Zielinski,[1] Bernhard O. Palsson[1,2,3,4,5]

**ABSTRACT** Fast growth phenotypes are achieved through optimal transcriptomic allocation, in which cells must balance tradeoffs in resource allocation between diverse functions. One such balance between stress readiness and unbridled growth in *E. coli* has been termed the fear versus greed (f/g) tradeoff. Two specific RNA polymerase (RNAP) mutations observed in adaptation to fast growth have been previously shown to affect the f/g tradeoff, suggesting that genetic adaptations may be primed to control f/g resource allocation. Here, we conduct a greatly expanded study of the genetic control of the f/g tradeoff across diverse conditions. We introduced 12 RNA polymerase (RNAP) mutations commonly acquired during adaptive laboratory evolution (ALE) and obtained expression profiles of each. We found that these single RNAP mutation strains resulted in large shifts in the f/g tradeoff primarily in the RpoS regulon and ribosomal genes, likely through modifying RNAP-DNA interactions. Two of these mutations additionally caused condition-specific transcriptional adaptations. While this tradeoff was previously characterized by the RpoS regulon and ribosomal expression, we find that the GAD regulon plays an important role in stress readiness and ppGpp in translation activity, expanding the scope of the tradeoff. A phylogenetic analysis found the greed-related genes of the tradeoff present in numerous bacterial species. The results suggest that the f/g tradeoff represents a general principle of transcriptome allocation in bacteria where small genetic changes can result in large phenotypic adaptations to growth conditions.

**IMPORTANCE** To increase growth, *E. coli* must raise ribosomal content at the expense of non-growth functions. Previous studies have linked RNAP mutations to this transcriptional shift and increased growth but were focused on only two mutations found in the protein's central region. RNAP mutations, however, commonly occur over a large structural range. To explore RNAP mutations' impact, we have introduced 12 RNAP mutations found in laboratory evolution experiments and obtained expression profiles of each. The mutations nearly universally increased growth rates by adjusting said tradeoff away from non-growth functions. In addition to this shift, a few caused condition-specific adaptations. We explored the prevalence of this tradeoff across phylogeny and found it to be a widespread and conserved trend among bacteria.

**KEYWORDS** transcriptional regulation, sigma factors, ribosomes, *Escherichia coli*

Maintaining optimal fitness in microorganisms requires navigating tradeoffs in resource allocation (1) due to dependencies between growth and expression (2–4). High growth rate expression states have been shown to downregulate stress response genes ("fearful" genes) and upregulate ribosomal genes ("greedy" genes) (1). Furthermore, this tradeoff has been well documented in adaptive laboratory evolution (ALE) experiments (5–8).

We have recently shown that transcriptional shifts of the *E. coli* transcriptome can be viewed through the use of a novel transcriptomic analysis method which uses

Address correspondence to Bernhard O. Palsson, palsson@ucsd.edu.

The authors declare no conflict of interest.

See the funding table on p. 15.

independent component analysis on large-scale expression databases to define sets of genes that are independently modulated, forming data-driven regulons termed iModulons (5). Through this analysis, we have identified that the fear versus greed (f/g) tradeoff is characterized by the strong negative correlation between the activity levels of the RpoS (fear) and Translation (greed) iModulons. The f/g tradeoff involves an upregulation of ribosomal genes (greed represented by the Translation iModulon) that often are the limiting factor for increasing growth rate (9) and a concurrent downregulation of stress-related genes (fear represented by the RpoS iModulon). While these iModulons and genes do not encompass all potential growth- and stress-related iModulons and genes within E. coli, they are unique in that they follow this tradeoff across a wide variety of conditions.

In addition to the two primary f/g iModulons, the GadX iModulon is also involved in the fear response, while the ppGpp iModulon adds another dimension to greed. The Translation iModulon primarily consists of ribosomal subunits, the RpoS iModulon contains the general stress response sigma factor RpoS's regulon, the GadX iModulon is related to acid stress, and the ppGpp iModulon is composed of genes involved in protein translation rates and the stringent response. The tradeoff between these sets of iModulons involves competition between the housekeeping and stress sigma factors (RpoD and (RpoS), binding of ppGpp and DksA to RNAP which modifies which genes RNAP transcribes, and other regulatory mechanisms (10–12). Many of these mechanisms directly involve RNA polymerase (RNAP) whose availability, along with sigma factor competition, has been previously connected to said tradeoff (13).

RNAP mutations have been shown to drive the f/g tradeoff toward faster growth, and RNAP is one of the most common mutation targets during ALEs (14). In a detailed study of two RNAP mutations found in the catalytic center, it was hypothesized these RNAP mutations adjust the tradeoff toward greed by destabilizing the rpoB-rpoC interface, thus affecting the binding of ppGpp to RNAP (15). While many ALE mutations cluster in the catalytic center of RNAP, there are numerous other RNAP mutations found in ALE endpoint strains. These mutations can be found near regulator binding sites, regions known to be related to antibiotic resistance, important structural elements such as the flap domain and trigger loop, and in regions with no clear annotations (16–24). Convergent RNAP mutations have been found in specific environmental adaptation experiments (25–28) often leading to the assumption that RNAP mutations reflect media adaptations, missing their underlying role in the f/g tradeoff. Despite being highly common evolutionary adaptations, the effect of these mutations is largely unknown.

Here, we sought to expand our knowledge of these RNAP mutations and the f/g tradeoff through a multi-scale study incorporating FAIR (Findable, Accessible, Interoperable, Reusable) data principles by using previously generated data and creating new easily accessible data (29). We first gathered the existing data on RNAP mutations and selected 12 mutations to address the shortcomings of said existing data. We then introduced the 12 RNAP mutations and used computer simulations to infer how these mutations destabilize RNAP. We then obtained transcriptomes in various experimental conditions and used iModulon analysis to demonstrate that, despite structurally distinct locations, these mutations nearly universally downregulate stress-related genes and upregulate growth-related genes (see Fig. S1) in addition to some condition-specific adaptations. We explored additional dimensionality of the tradeoff involving the ppGpp and GadX iModulons. Finally, we compared the transcriptomes of various species to find that f/g tradeoff is widely found across phylogeny. Thus, our multi-scale study elucidated key features of a central transcriptomic tradeoff between fear and greed in which cells that favor faster growth face the cost of diminished responsiveness to stresses (15) and proposed that it is a general principle in microbiology.

## RESULTS

### Creation of a new data set of common RNAP mutations

RNAP mutations are frequently fixed in ALEs, with 36% of evolved isolates in ALEdb, a database of mutations acquired during ALE (14), containing at least one RNAP mutation: 6% have a *rpoA* mutation, 20% have a *rpoB* mutation, and 13% have a *rpoC* mutation. For this study, 12 RNAP mutations were selected and generated for experimental evaluation using three primary criteria: (i) the frequency of occurrence of the mutation in *E. coli* ALE endpoints, (ii) their structural location in relation to a known RNAP region of interest, such as effector binding sites, and (iii) evidence of phenotypic impact of the mutation. The 12 chosen mutations and their characteristics are summarized in Table S1. Figure 1A shows the location of these 12 mutations on RNAP along with some particular structural regions of interest. Sigma factor binding sites are shown in Fig. S2. These mutations were introduced into the genome of the model K-12 MG1655 strain of *E. coli* (see Materials and Methods, "Creation of RNAP mutations") to generate single mutation knock-in strains.

RNA-sequencing data were collected under aerobic growth on glucose M9 minimal media for each of these individual mutants (see Materials and Methods, "RNA-sequencing"). Some of the RNAP mutant strains were additionally tested under specific stress conditions that were similar to the ALE experiment in which they were originally found (see Table S2). All but one of the 12 mutants exhibited a shift toward greed in the f/g tradeoff in the transcriptome (Fig. 1B). The exception, *rpoB* I966S, arose during an evolution to high-temperature growth (32) and may, therefore, have had a stronger impact on temperature stability than regulation of expression. All but *rpoB* I966S and *rpoC* N309Y, the latter of which arose during butanediol tolerance evolutions, increased the growth rate (Fig. 2A). *RpoC* N309Y does not increase the growth rate but does shift its transcriptome toward greed in a pattern consistent with the other mutations (Fig. S3). It should be noted that *rpoC* N309Y was generated using a different procedure from the other mutations (see Materials and Methods, "Creation of RNAP mutations") which could be skewing its results.

### RNAP mutations destabilize the rpoB-rpoC interface and likely affect sigma factor binding

RNAP mutations have been shown to affect RNAP structurally in a variety of ways. Some of the most commonly found and widespread RNAP mutations are *rpoB* E672K, *rpoB* P1100Q, *rpoB* G1189C, and *rpoC* N720H (Fig. 2). The physical mechanism for how these four mutations cause the tradeoff is not fully established, but some key properties are known. Structurally, they are all located near the rpoB-rpoC interface (*rpoB* E672K = 5.46 Å, *rpoB* P1100Q = 5.24 Å, *rpoB* G1189C = 8.97 Å, *rpoC* N720H = 10.09 Å) as visualized in Fig. 2B. PyRosetta (31) was used to calculate the mean impact of these mutations on the holoenzyme structures (see Table S3) and found that all were predicted to destabilize the rpoB-rpoC interface (*rpoB* E672K = −28.40 REU, *rpoB* P1100Q = −23.98 REU, *rpoB* G1189C = −5.26 REU, *rpoC* 1055V = −13.91 REU, mean of all RNAP mutations on ALEdb = −16.83 REU, see Fig. S4). This region is nearby to a ppGpp-binding site which the mutations are also mostly predicted to destabilize (see Table S1) and, thus, likely modify its regulatory role (19) which is tightly connected to RpoS's own activity (33). The effect these mutations have on RNAP though are unlikely only limited to the destabilization of said interfaces.

These mutations each may have effects specific to their structural location. *RpoB* E672K, for example, is located at the base of the bridge helix where it possibly affects DNA-RNAP interactions. *RpoB* P1100Q is near a helix in the beta prime subunit that interacts with ppGpp-binding site 1. A more thorough description of each mutation's possible specific structural impacts is available in the supplement (see Table S4). While some of these mutations are near to ppGpp-binding site 1, it should be noted that ppGpp-binding site 2 has been reported to have a greater effect on gene expression (19).

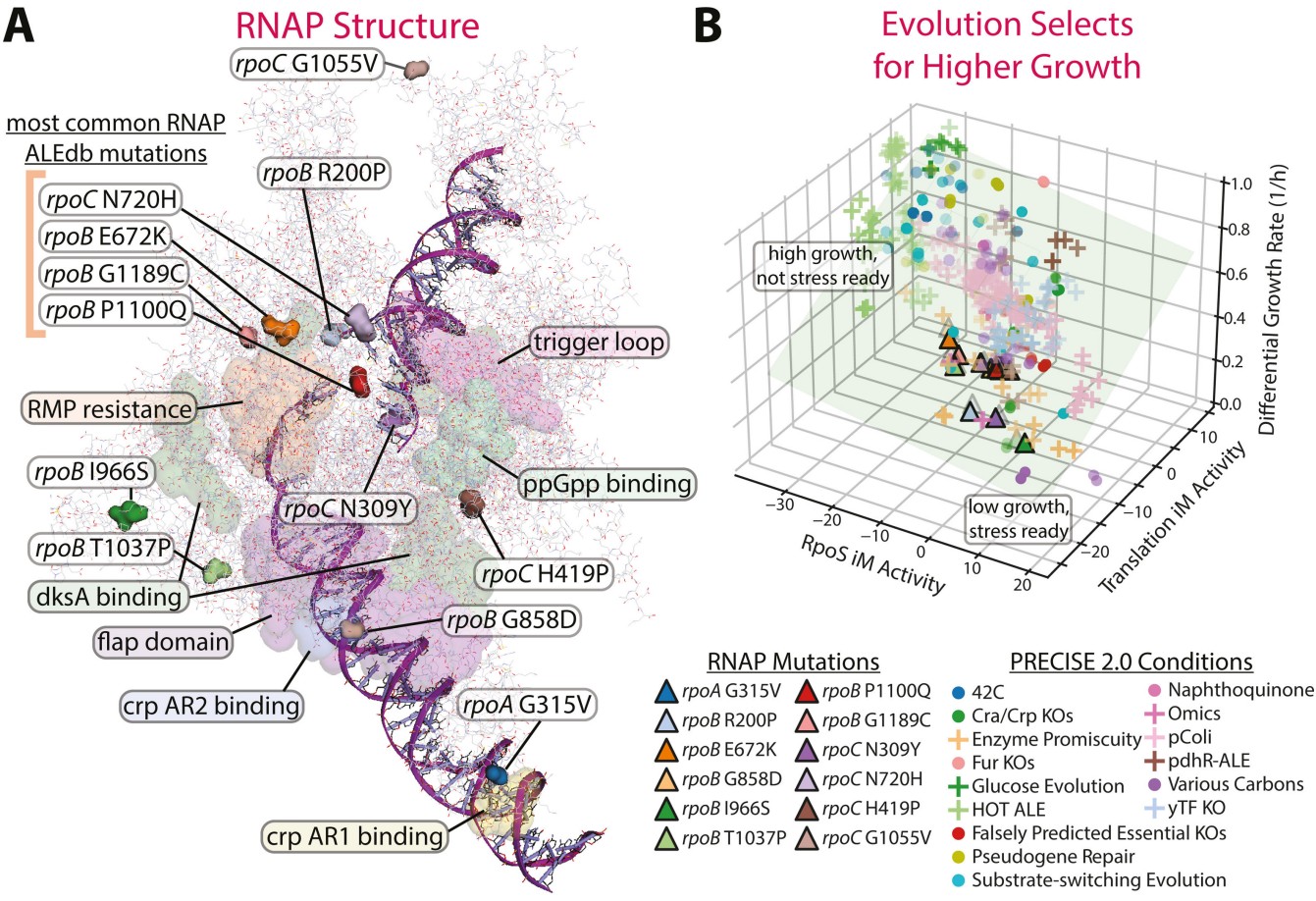

**FIG 1** RNAP mutations alter the fear vs greed tradeoff. (A) The structure of RNAP (PDB 6OUL [30]) is visualized using PyRosetta (31), showing the location of mutations used in this study and highlighting some specific RNAP regions of interest (16–24). The grouped mutations on the upper left are some of the most common mutations found in ALEdb (14) and are further discussed in Fig. 2. (B) Laboratory evolution leads to sequence variants which adjust the composition of the transcriptome leading to faster growth and repressed stress readiness. The f/g tradeoff on the transcriptome is shown (RpoS represents fear, Translation represents greed) along with the mutations' impact on growth rate. All PRECISE 2.0 samples with recorded growth rates are shown. Growth rates are centered on their respective experiments' unevolved control conditions. The green plane is fitted to the data and shows that growth rates increase with lower RpoS and higher translation iModulon activities.

DksA, which comprises much of the interface with ppGpp in site 2, has not been mutated in samples found in ALEdb.

Unfortunately, a computational structural analysis of how these mutations affect sigma factor binding is not feasible. Sigma factors bind over large portions of RNAP (see Fig. S2) and the specific structural file used has a dominant effect on the resulting destabilization scores. What we can observe though is that the RNAP mutations modify RNAP's interactions with sigma factors nonuniformly. Genes regulated by RpoS (34), the general stress response sigma factor, showed on average a −0.33 change in $\log_2$ transcripts per million (tpm) expression when compared to the wild-type. The relatively small change (−0.059 change in $\log_2$ tpm) in genes regulated by RpoD, the housekeeping sigma factor, shows that these mutations differentially affect sigma factor functions (see Table S5 for average change of all sigma factors). This infers that these mutations are preferentially affecting certain sigma factors likely through their binding interfaces.

## RNAP mutations lead to upregulation of growth-related genes and downregulation of stress-related genes

The analysis of global changes in the transcriptome is difficult due to the high number of differentially expressed genes in many comparisons. Furthermore, comparing many

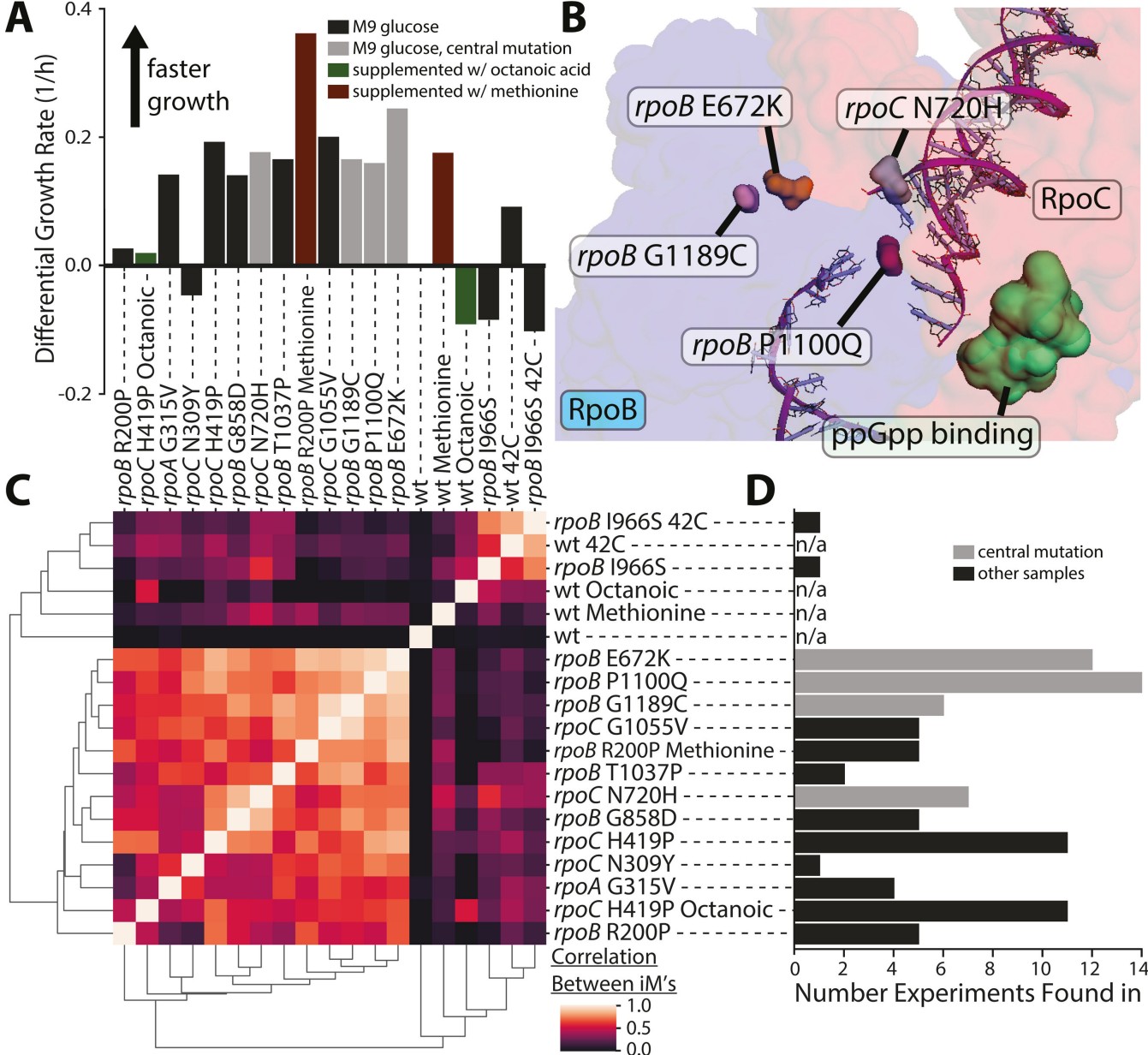

**FIG 2** Transcriptome similarities and location of common ALE-acquired RNAP mutations. (A) The growth rates of the mutated strains relative to the wild-type control. (B) A subsection of RNAP (PDB 6OUL [30]) showing the location of common mutations with respect to the rpoB-rpoC interface and the ppGpp-binding site, visualized using PyRosetta (31). (C) Correlations between the activity levels of all iModulons between RNAP mutants under the same growth condition. The plot shows that all the 12 mutations have a similar impact on transcriptome composition. Mutations in the catalytic core have a near-identical impact on the transcriptome. (D) Number of laboratory evolution experiments that RNAP mutations are fixed in (number given is from a total of 743 ALE experiments found in ALEdb [14]). The gray bars in this panel and panel C are the mutations grouped as "most common RNAP ALEdb mutations" in Fig. 1 and are visualized on the RNAP structure in panel B of this figure (31).

conditions is challenging if pairwise differential expression of genes (DEG) plots are used (35) (see Fig. S5). To overcome these challenges, we used the iModulon workflow (5, 36) to identify independently modulated gene sets (iModulons) and interpret their differential activity between all conditions used. This workflow uses independent component analysis (ICA) of a compendium (X) of RNA-sequencing data, which includes our samples of interest along with a variety of other experiments which help to separate source signals associated with transcriptional regulators (5, 36). The algorithm generates two output matrices: M (whose columns highlight the genes in each iModulon) and A (whose

rows show the iModulon's activity in every sample). Detailed information on each iModulon is available at iModulonDB.org (37), and this study focuses primarily on the "*E. coli* PRECISE 2.0" data set (36), an *E. coli* database of RNA-sequencing data obtained under 422 growth conditions. All iModulon activities are measured relative to an unstressed M9 glucose condition and should be interpreted thusly.

Principal component analysis (PCA) of the iModulon activity matrix (A) shows that much of its variance and, thus, expression variation, in general, is explained by the RpoS and Translation iModulons' activities (see Table S6). The RpoS iModulon is the largest and the Translation iModulon is the fifth largest contributing factor to the highest variance explaining principal component (PC). GadX and ppGpp iModulons are also highly contributing factors to large variance explaining PCs, adding additional dimensionality to f/g that is further explored in Fig. 3. The f/g tradeoff is, thus, a major contributor to variation in the composition of the transcriptome.

The new RNA-sequencing data from the 12 new RNAP mutant strains were analyzed using ICA (5). The iModulon activity levels in the new samples were compared to those in PRECISE 2.0. This database was used to compute the iModulons structure of the *E. coli* transcriptome (5), and the gene composition of the key fear and greed iModulons is found in Table S7. Links to the iModulonDB pages for each are provided in said table where the overlap between regulators and iModulon genes along with gene annotations can be easily viewed.

All of the 12 mutations introduced, except for *rpoB* I966S, have a large impact on the activity level of the RpoS iModulon similar to the two previously studied RNAP mutations (15). The mutations in the catalytic center (those visualized in Fig. 2B) have the largest impact on RpoS iModulon activity levels (44.1% higher on average than the other RNAP mutants generated for this study as can be seen in Fig. S3), but mutations distant from this location can also strongly impact the activity of this iModulon which has not been previously shown. This suggests there is more complexity to the physical mechanism of this transcriptomic effect. Both the frequency of occurrence and the effect of these RNAP mutations found in the catalytic center imply they are commonly fixed during growth rate selection (i.e., maximization of "greed").

## Genome-scale models of proteome allocation quantitatively estimate the growth benefit of maximizing greed functions

While iModulons are an informative approach to reveal the hallmarks of changes in the expression state, they are not directly representative of the composition of the proteome. Creating iModulons from expression data requires the input RNA-sequencing data to be both centered to a control and normalized. This means the activity levels of iModulons for samples are entirely relative to each other, and their magnitude range is constrained by the variance of the PRECISE data set. We, thus, deployed a genome-scale model to reproduce the f/g tradeoff which allowed us to infer absolute measures of the proteome of cells undergoing said tradeoff. A genome-scale metabolism and expression (ME) model (38) was run to maximize growth while constraining RpoS iModulon-associated reactions to a specified lower bound.

The resulting RpoS and Translation iModulons' proteomic computed mass fractions were highly anticorrelated (−0.9994) (see Fig. S6). A unit activity increase in the Translation iModulon has a 650% stronger effect on said iModulons' genes' proteome mass fraction than it does in the RpoS iModulon (see Materials and Methods). This implies that the small activity increases of the Translation iModulon seen in the f/g tradeoff and, in the RNAP mutations, may be having a larger effect than appears on the cell's phenotype. This computational model also indicates that forced expression of the stress readiness genes reduces the expression of the growth-promoting genes as experimentally observed.

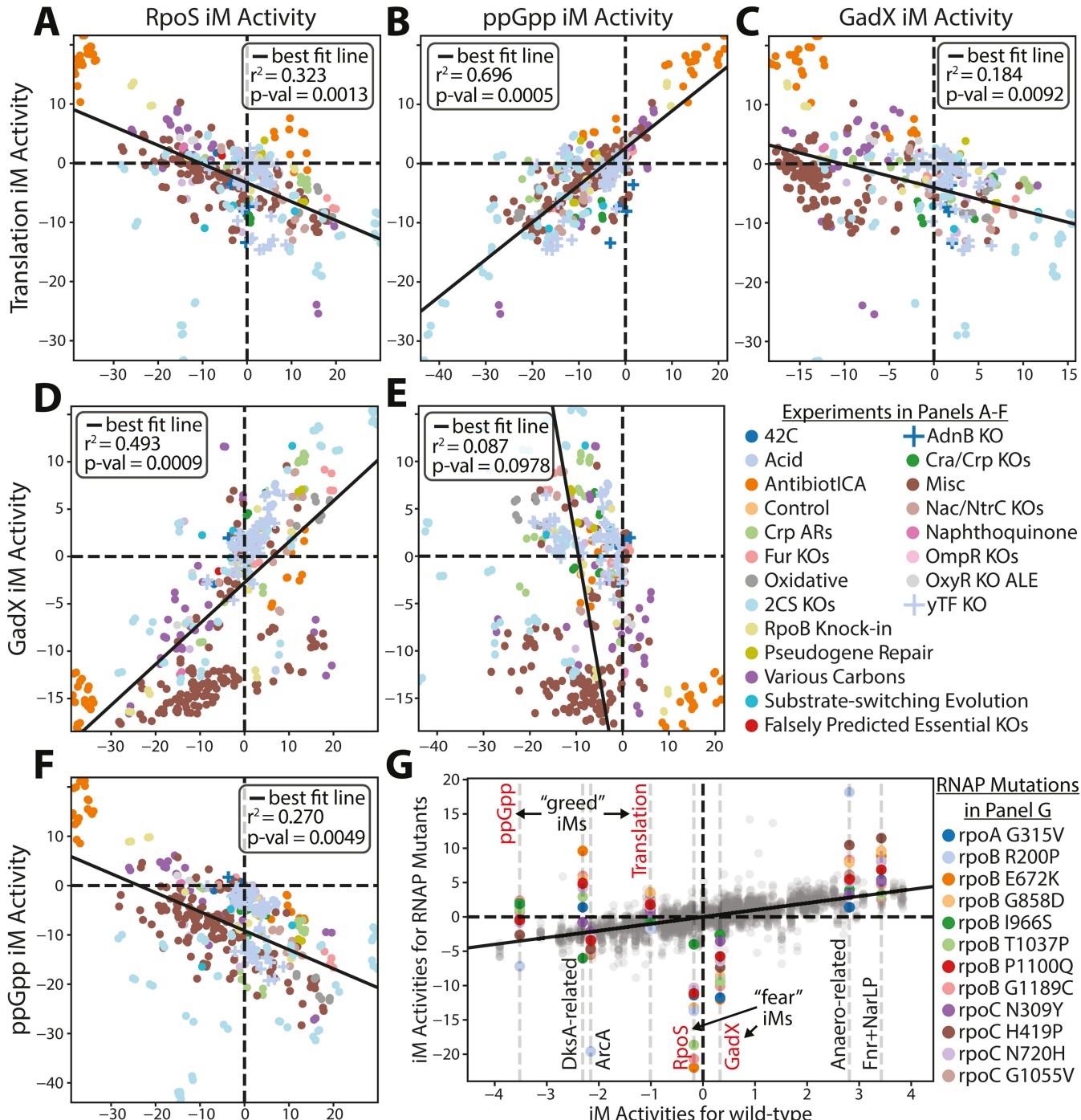

**FIG 3** Reflections of the fear vs greed tradeoff transcriptome in the relative activity levels of the translation and stress iModulons. (A–F) These plots show the relationship in activity levels between the greed (Translation and ppGpp) and fear (RpoS and GadX) iModulons. The *P*-value is calculated using a t-distribution test of all iModuon-to-iModulon pairwise activity level comparisons. (G) The activity levels of various growth- and stress-related iModulons for the RNAP mutants, along with some other iModulons highly affected by said mutations. The gray dots are the activity levels of the other iModulons for all of the mutants. Red labeled iModulons are plotted in panels A–F.

## The fear vs greed tradeoff additionally involves GAD and ppGpp iModulons

The f/g tradeoff was first visualized using the activity levels of the Translation and RpoS iModulons (5). Since this study was published, the number of transcriptomes for *E. coli* has quadrupled (35). The analysis of the larger data sets reveals additional

dimensionalities to the f/g tradeoff. Several additional iModulon activity levels are correlated with growth rates, including the GadX and ppGpp iModulons. The RNAP mutations are likely affecting these iModulons as ppGpp binds to RNAP, while the GAD regulon's expression has been closely tied to RpoS and ppGpp (10, 39). GadX is highly correlated with RpoS (0.71) and negatively correlated with growth rates (−0.24), while ppGpp is strongly correlated with Translation (0.74) and has a weak positive correlation with growth rates (0.14). Correlation plots for each of these iModulon activity pairings are given in Fig. 3A through F. All pairings except for GadX and ppGpp iModulons show a clear correlation.

## RNAP mutations can be condition-specific adaptations

While the core group of common RNAP mutations downregulate stress-related iModulons and upregulate growth-related iModulons (Fig. 3G), other RNAP mutations have more specific effects that are adaptations to the environments from which they were found. Figure S7 shows two of these such mutations (*rpoB* R200P and *rpoA* G315V) from our set of 12 mutations.

The *rpoB* R200P mutation reflects a specific selection condition. It is found commonly in replicate methionine tolerance evolutions (27), and it has two effects on the transcriptome: (i) during growth on methionine, it activates the Translation iModulon and downregulates the RpoS iModulon to increase the growth rate compared to wild-type and (ii) during growth on M9 glucose, it activates anaerobic response genes found in the Fnr-1, Fnr-2, and Anaero-related iModulons. These responses are likely used because methionine contains sulfur and is, thus, a common target of reactive oxygen species (ROS) in *E. coli* (40).

The *rpoA* G315V mutation affects the activities of Crp-1 and Crp-2 iModulons with the strongest impact on the maltose operons. This mutation was found in a *pgi* synthetic gene replacement ALE (26) in nearly all strains that failed to integrate the exogenous *pgi* replacements. Presumably, the loss of *pgi* required large changes to sugar import systems, thus necessitating this *rpoA* mutation to help downregulate maltose importers (41). The mutation's effect on the Crp-1 iModulon is similar to one reported in a study that deactivated regions of *crp* (42) (see Fig. S7E). Thus, it is likely the mechanism of action for this *rpoA* mutation is to modify the rpoA-crp binding interface.

Thus, there are RNAP mutations outside the core of the enzyme that confer condition-specific effects on the transcriptome (see Supplemental Information for more cases). This observation leads to a wider examination of the effects of RNAP mutations that are selected under specific conditions.

## The genetic basis for the fear vs greed tradeoff during ALE is condition-dependent

The primary fear and greed iModulons are correlated for both unevolved samples (−0.57 correlation) and evolved samples (−0.39 correlation, see Fig. 4A) although evolved samples strongly favor greed. These correlations hold true across samples with and without RNAP mutations, but RNAP mutations nearly universally favor a movement toward greed. Different stressors lead to specific transcriptional adjustments along the f/g tradeoff to best favor growth as is annotated in Fig. 4A.

In most laboratory evolutions with high stress conditions, evolution downregulates the RpoS iModulon over time. The cells initially use the RpoS iModulon to respond to nearly any stress but eventually tune the stress response to the specific environment. In a reaction oxygen species experiment (labeled ROS TALE) (43), initially, the RpoS iModulon was highly active, but as the cells evolved on paraquat most of the iModulon was downregulated, while the expression of oxidative response genes in the iModulon was left largely unmodified (see Fig. S8). This transcriptional regulatory network adjustment, which was driven by convergent mutations to *icd, aceE, sucA, oppA,* and *emrE* among other genes, enabled the cells to grow faster in a ROS stress environment.

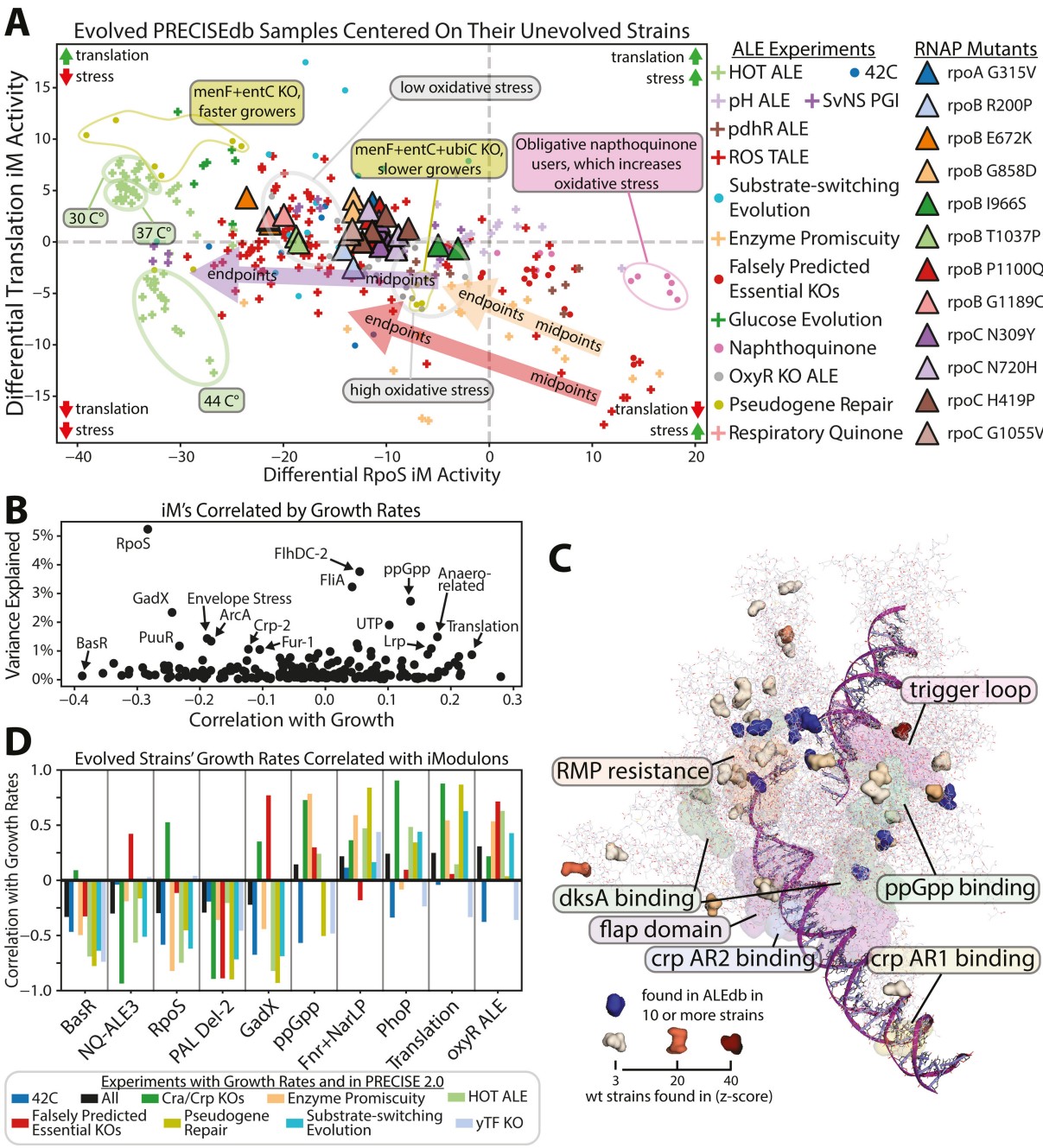

**FIG 4** Fear vs greed adaptations vary between laboratory and natural conditions. (A) The fear vs greed iModulon activities of the laboratory-evolved samples of PRECISE 2.0 centered on their respective unevolved wild-type strains' iModulon activities. The triangles are the impact of the individual RNAP mutations from Fig. 1. Specific annotations are given for different adaptation experiments showing variations in the effects of the selection pressure. Where midpoint strains are available, arrows indicate evolution from midpoint to endpoint strains. General effects of each quadrant are summarized in their respective corners. Details of these experiments are available in Table S8. (B) The iModulons of PRECISE 2.0 correlated with the available growth data along with how much of the total transcriptome's variance they explain. (C) The most common mutations found in ALEdb and the natural variants of RNAP (PDB 6OUL [30]) visualized using PyRosetta (31). (D) The correlation values between growth rates and iModulons for all evolution experiments with growth rates reported.

## Growth rates are well correlated with the fear vs greed tradeoff

Standardizing growth rates across experiments is a difficult task, as unintentional differences in laboratory procedures, data processing, or simple measurement bias can drastically skew the results while intentional differences in experimental conditions make

a direct comparison difficult. Growth rate data are also often not reported, as just 43% of PRECISE 2.0 samples have associated growth rates. The growth data present, however, support the fear vs greed tradeoff. Translation is the 2nd most positively correlated iModulon with growth, while the RpoS iModulon is the 10th most negatively correlated with growth (Fig. 4B). It is important to note that the iModulons with stronger correlations to growth than Translation and RpoS explain little of the transcriptome's variance. Figure 4D shows that these correlations hold true across a variety of evolution experiments.

OxyR ALE, for example, is the most correlated positively iModulon with growth rates yet explains only 0.1% of the transcritome's variance and its activity is nearly entirely limited to the ALE study for which it is named. Anaero-related, in addition to Translation and ppGpp, has a positive correlation with growth and a large explained variance of PRECISE's expression data. While it also is upregulated by the RNAP mutations, compared to the other greed iModulons, it contains many genes of unknown function and has no clear regulator. The f/g tradeoff is defined not by all growth and stress-related genes but rather key well-defined stress and growth iModulons whose activities anti-correlate with each other over a large range of conditions. However, future versions of PRECISE will likely enable the inclusion of the Anaero-related iModulon among others into the f/g tradeoff.

This ceaseless pull toward greed and away from stress readiness, however, is largely limited to laboratory conditions. The lack of overlap between the natural variants (44) and the ALEdb mutations is seen in Fig. 4C implies that there are highly divergent evolutionary pressures on wild-type strains and their ALE counterparts.

## The fear vs greed tradeoff is found across the phylogenetic tree

Finally, we searched the phylogenetic tree for other organisms exhibiting the f/g tradeoff (the phylogenetic tree highlighting said species can be seen in Fig. S9). First, we analyzed data from a multi-strain *E. coli* ALE study (45). This analysis shows that the tradeoff was found in all the *E. coli* strains of the study (Fig. 5A). Second, we examined iModulonDB (37) for the presence of the f/g in other species (Fig. 5D through K). The tradeoff was clearly found in 7 out of the 12 bacterial strains surveyed (see Materials and Methods, "Cross-species iModulon comparisons"). Although the gene composition of the fear iModulons varies between species (likely a consequence of differing stresses in their natural environments), all of the primary greed iModulons consist of a highly similar set of ribosomal subunits and translational associated functions (Fig. 5B and C). The five species in which the tradeoff was not found all contain a greed iModulon that consists primarily of ribosomal subunits, but said species contain no one clear stress iModulon that correlates with it. The presence of the greed-related genes of the f/g tradeoff across such a wide range of species implies that they may be a global property of bacterial transcriptomes.

## DISCUSSION

We detail a general tradeoff in the bacterial transcriptome between growth rate and stress readiness. A major genetic component of this tradeoff lies in RNAP mutations, which affect the structure of RNAP and, consequently, the composition of the transcriptome. In RNAP mutants that arise from ALE studies, the modified transcriptome composition favors the transcription of growth-related functions over stress-related functions. The tradeoff between fear and greed-related functions was found across a wide range of wild-type bacterial strains. Similar transcriptional tradeoffs have been seen before in persistence (47), nutritional competence (48), and protein cost in metabolic pathways (49). Interestingly, the fear vs greed tradeoff has been described in many areas of science, such as economics (50), game theory (51), and psychology (52). It has been elucidated here for microbiology through a multi-scale analysis.

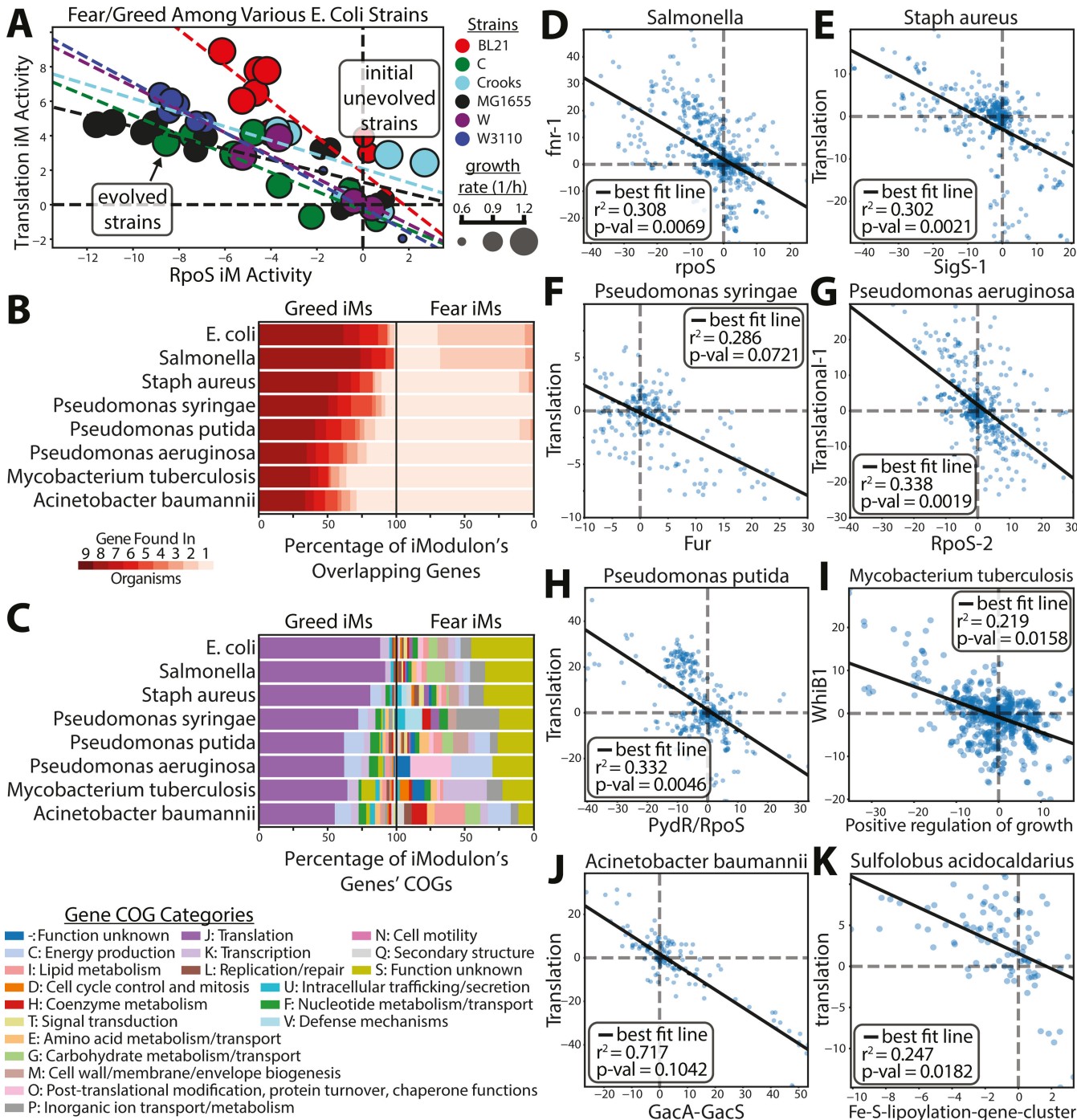

FIG 5 The fear vs greed tradeoff is found across the phylogenetic tree. (A) The f/g tradeoff appears in ALEs across multiple *E. coli* strains (46). (B) Percentage of genes found in common among translation and stress iModulons in different species. Many genes are commonly found in the Translation iModulons, while the genes found in the fear iModulons are more disparate. See Materials and Methods for a more detailed description of the iModulon comparison methodology. (C) The COG category of the genes of the greed and fear iModulons. (D–K) The f/g tradeoff among a variety of species found in iModulonDB (37). The *P*-value is calculated using a *t*-distribution test of all iModuon-to-iModulon pairwise activity level comparisons. The names of the iModulons are pulled from their respective data sets. The *Mycobacterium tuberculosis* "Positive regulation of growth" iModulon mostly consists of stress-related antitoxin genes.

A previous study compared two RNAP mutations (15), *rpoB* E672K and *rpoB* E546V, and found that they destabilize the rpoB-rpoC interface (53). Another study using *in vitro* assays linked an *rpoC* deletion from 3,611 to 3,619 bp (near to the rpoB-rpoC interface) to

destabilizing the open complex of RNAP which led to decreased transcriptional pausing on the promoter, reduced RNAP's open complex half-life, and increased elongation rates (54). For our centrally located mutations, our evidence best supports this model of a destabilized rpoB-rpoC interface leading to a destabilized open complex, thus, causing transcriptional changes. However, we have no clear mechanistic explanation as to why mutations distant from this central region, such as *rpoC* G1055V, have similar impacts to the transcriptome. The impact of RNAP mutations has also been shown to be similar to strains with a reduced number of ribosomal operons, suggesting that these mutations are possibly modifying ribosomal availability and/or distribution (55). Other RNAP mutations were found to eliminate the destabilizing effect of ppGpp binding to RNAP, thus reducing the inhibition of transcription by ppGpp (56).

A recent study analyzing 45,000 ALE mutations and comparing them to wild-type variant alleles suggests that under laboratory evolution the wild-type alleles are under negative selection pressure, while ALE mutations are under positive selection pressure (44). This suggests that ALE mutations represent extreme mutations extenuating a preferred trait, thus amplifying the basis for the f/g tradeoff as opposed to nature in which a sole focus towards faster growth would leave cells unable to adapt to highly variable conditions.

The current study expanded upon current knowledge (15, 54) by analyzing the impact of 12 RNAP mutations to detail RNAP's role as a global master regulator of the f/g tradeoff. All 12 of these mutations, however, are from evolution experiments, and their common adjustments toward greed are reflective of that. The detailed molecular/structural mechanisms that underlie the tradeoff are not fully understood but appear to involve the rpoB-rpoC interface (15) and other important structural regions of RNAP, altered kinetic and regulatory properties (54), and changes in the sigma factor use of RNAP.

The effects that RNAP mutations have on the transcriptome composition, however, are clear. The transcriptomic re-allocation involves a consistent set of iModulons with known functions. As additional versions of PRECISE are created using more data, it is likely additional iModulons could be included in this tradeoff. The relationship between the proteome and transcriptome functions enables genome-scale computational biology assessment of the phenotypic consequences of the reallocation (57). Thus, a detailed understanding of the effects of the f/g tradeoff at the systems level has emerged. As the tradeoff involves resource allocations for improved fitness, it is important to contextualize particular RNAP mutations fixed in laboratory evolution studies and seek to identify adaptive mutations that are condition specific.

Finally, the phylogenetic distribution of the greed-related genes of the f/g tradeoff is broad, suggesting that this tradeoff may emerge as a universal feature of the bacterial transcriptome that can be captured by iModulons. It is not known, however, if RNAP mutations would have a similar impact on the tradeoff in these species. The tradeoff has been also found in a minimal synthetic organism, further supporting its potential ancient origin (58). RNAP and the f/g tradeoff have been shown to play a highly important role in balancing growth and stress adaptations.

## MATERIALS AND METHODS

### Strain information

*E. coli* K-12 MG1655 was used as the wild-type and as the source strain for all mutations created for this study.

### Creation of RNAP mutations

*RpoC* N309Y was created using pORTMAGE (59), the protocol for which is included in File S1. Initially, pORTMAGE was intended to be used to generate all strains, but only *rpoC* N309Y could successfully be generated, and thus, the rest were created using the

CRISPR-based protocol outlined in Zhao et al. (60). Mutations were verified using reverse PCR. Primer sequences used in the generation of mutants are included in File S2.

## Growth rate calculations and comparisons

Reproductive growth rates were calculated under the same conditions for all RNAP mutated strains. A 24-well magnetic heat lock set to 37°C was used for continual cultures. Sixteen milliliters culture with $OD_{600} = 0.05$ using M9 minimal media supplemented with 4 g/L glucose was prepared in a plastic tube, and time points were taken in replicate for growth rate calculation approximately every 30 minutes.

For comparing growth rates across experiments, all growth rates were analyzed as differential values relative to their respective experiment's control condition. After being centered on their respective control conditions, the differential growth rates were normalized for each experiment. The growth rate values for the PRECISE 2.0 samples are available at iModulonDB (https://imodulondb.org/organisms/e_coli/precise2/data_files/sample_table.csv).

## RNA-sequencing

All samples were prepared and collected in biological duplicates. Three millilters of culture isolated at an $OD_{600}$ of 0.5 was added to 6 mL of Qiagen RNA-protect Bacteria Reagent after sample collection. This solution was then vortexed for 5 seconds, incubated at room temperature for 5 minutes, and then centrifuged. The supernatant was then removed, and the cell pellet was stored at −80°C. The Zymo Research Quick-RNA MicroPrep Kit was used to extract RNA from the cell pellets per vendor protocol. On-columns DNase treatment was performed for 30 minutes at room temperature. Anti-rRNA DNA Oligo mix and Hybridase Thermostable RNase H (61) were used to remove ribosomal RNA. Sequencing libraries were created using a Kapa Biosystems RNA HyperPrep per vendor protocol. RNA-sequencing reads were processed using https://github.com/avsastry/modulome-workflow. Data are available at NCBI GEO GSE227624.

## iModulon computation

RNA-sequencing data were used to create iModulon activity levels of the mutated strains using PyModulon (5) which is available at https://github.com/SBRG/pymodulon. Activities of iModulons were compared to samples from PRECISE 2.0 (36) which is easily accessible using iModulonDB (37).

## Mutation analysis

ALEdb (14) was used for selecting the mutations for this study. Any *E. coli* strains on ALEdb were considered potential sources for mutations. Mutations from the same sample but where one is from an isolate and one is from the population were considered to be just one instance of said mutation.

## Structural analysis

Structural analysis was performed using PyRosetta (31) using its default score function. The pdb files were downloaded from RCSB (62). REU stands for Rosetta Energy Unit, which is PyRosetta's unit for energy. Structural files used for the analysis are shown in Table S3. These files were selected primarily based on a review of bacterial RNAP (63).

The structural calculations for the rpoB-rpoC-binding interface were performed by calculating the binding energy between the chains coded by *rpoB* and *rpoC* using the holoenzyme pdb structures. For each mutation, said mutation was introduced, the protein was repacked, and the binding energy between the two chains was recalculated and compared to the baseline. The structural calculations for ppGpp-binding analyses were carried out similar to the rpoB-rpoC-binding simulations, but by instead calculating the binding energy between ppGpp and the rest of the protein.

Structural analysis was carried out for each of the 12 mutations created specifically in this study (see Fig. S4). Calculations were also carried out for an alanine scan of RNAP and all ALEdb RNAP mutations to serve as various controls (see Fig. S10).

## Data processing

iModulons are calculated using expression data centered on a control. For this study, the control was a wild-type M9 glucose growth sample on which all other samples were centered. All biological replicates of expression data had over 99% correlation to each other and were averaged together. In addition to PyRosetta (31) and PyModulon (5), numpy (64), pandas (65), and scipy (66) were used to generate figures and perform analysis.

## Metabolic model and proteomic calculations

The FoldME (38) model was used for the metabolic modeling calculations. Figure S6 was generated by iteratively increasing the lower bounds for the genes of the RpoS iModulon and recording the proteomic mass fraction of the Translation and RpoS iModulons' genes until the model no longer ran. Proteome mass fraction to iModulon genes is the sum of the measured proteomic mass fractions of each enriched gene in an iModulon. This value is calculated for every sample and plotted against its corresponding PRECISE iModulon Activity. The proteomic calculations performed for this paper are well described in Patel et al. (57).

## Cross-species iModulon comparisons

To compare iModulons across different species, first genes from the various strains were matched to each other using Orthofinder on its default settings (67). The FASTA files for each organism were pulled from their respective NCBI genome pages and fed into the algorithm. The many-to-many Orthofinder results were used to generate the gene mapping for later steps. In the case that an organism had multiple genes mapped to one orthogroup, the multiple genes' weightings were averaged when mapped to the orthogroup. The many-to-many results were used based on the rarity of one-to-one orthologs and at the suggestion of Orthofinder's GitHub page.

Species' iModulons were mapped to both *E. coli*'s Translation and RpoS iModulons based on Fisher's exact test $P$-values generated on orthogroup presence/absence in iModulons. Said presence/absence calls were generated using $k$-means clustering of the orthogroup activity levels within iModulons with the number of clusters set to 2 and taking the smaller cluster as the orthogroups present in an iModulon. The iModulon from each species with the lowest $P$-values were selected as the best matching iModulon. In the case where no iModulon matched the *E. coli* RpoS iModulon (namely for *Mycobacterium tuberculosis* and *Acinetobacter baumannii*), the iModulon most negatively correlated with their Translation iModulon was chosen. iModulon names were pulled from the individual species' iModulons.

## ACKNOWLEDGMENTS

We would like to thank Marc Abrams (Systems Biology Research Group, University of California San Diego) for assistance with paper editing, José Utrilla (Center for Genomic Sciences, UNAM) for help with historical context, and Hyungyu Lim (Inha University) for generating the stationary phase data used in Fig. S11.

This work was supported by The Novo Nordisk Foundation (NNF) Center for Biosustainability (CfB) at the Technical University of Denmark (NNF20CC0035580), National Institutes of Health Grant R01 GM057089, and the Y. C. Fung Endowed Chair in Bioengineering at the University of California San Diego.

C.D and B.O.P. conceived and supervised the study. R.S., Y.H., and C.D. performed the experiments. A.P. carried out the genome-scale model simulations. C.D., K.R., and D.C.Z. performed the remaining analyses. All authors read and approved the final manuscript.

## AUTHOR AFFILIATIONS

[1]Department of Bioengineering, University of California San Diego, La Jolla, USA
[2]Bioinformatics and Systems Biology Program, University of California San Diego, La Jolla, USA
[3]Department of Pediatrics, University of California San Diego, La Jolla, California, USA
[4]Center for Microbiome Innovation, University of California San Diego, La Jolla, California, USA
[5]Novo Nordisk Foundation Center for Biosustainability, Technical University of Denmark, Lyngby, Denmark

## AUTHOR ORCIDs

Christopher Dalldorf http://orcid.org/0009-0004-9522-9648
Bernhard O. Palsson http://orcid.org/0000-0003-2357-6785

## FUNDING

| Funder | Grant(s) | Author(s) |
|---|---|---|
| Novo Nordisk Fonden (NNF) | NNF20CC0035580 | Christopher Dalldorf |
| | | Kevin Rychel |
| | | Arjun Patel |
| | | Daniel C. Zielinski |
| | | Bernhard O. Palsson |
| HHS \| NIH \| OSC \| Common Fund (NIH Common Fund) | GM057089 | Christopher Dalldorf |
| | | Kevin Rychel |
| | | Richard Szubin |
| | | Ying Hefner |
| | | Arjun Patel |
| | | Daniel C. Zielinski |
| | | Bernhard O. Palsson |
| Y.C. Fung Endowed Chair at UCSD | | Christopher Dalldorf |
| | | Kevin Rychel |
| | | Richard Szubin |
| | | Ying Hefner |
| | | Arjun Patel |
| | | Daniel C. Zielinski |
| | | Bernhard O. Palsson |

## DATA AVAILABILITY

Data are available at NCBI GEO GSE227624.

## ADDITIONAL FILES

The following material is available online.

### Supplemental Material

**Supplemental Material (mSystems00305-24-s0001.docx).** Supplemental text, tables, and figures.

## Open Peer Review

**PEER REVIEW HISTORY (review-history.pdf).** An accounting of the reviewer comments and feedback.

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
