## [Reviewer comments · mSystems]

The hallmarks of a tradeoff in transcriptomes that balances stress and growth functions

Christopher Dalldorf, Kevin Rychel, Richard Szubin, Ying Hefner, Arjun Patel, Daniel Zielinski, and Bernhard Palsson

Corresponding Author(s): Bernhard Palsson, University of California San Diego Jacobs School of Engineering

Review Timeline:

Submission Date:	February 29, 2024
Editorial Decision:	March 31, 2024
Revision Received:	April 11, 2024
Accepted:	April 24, 2024

Editor: Anthony Fodor

Reviewer(s): Disclosure of reviewer identity is with reference to reviewer comments included in decision letter(s). The following individuals involved in review of your submission have agreed to reveal their identity: Way Sung (Reviewer #2)

Transaction Report:

DOI: <https://doi.org/10.1128/msystems.00305-24>

Re: mSystems00305-24 (The hallmarks of a tradeoff in transcriptomes that balances stress and growth functions)

Dear Prof. Bernhard O Palsson:

Your paper has been reviewed again by our three external reviewers and there is consensus that the paper is acceptable for publication provided that the mostly minor comments they raise can be addressed. Please return the revised manuscript within 60 days and I should be able to approve it for publication without further external review; if you cannot complete the modification within this time period, please contact me. If you do not wish to modify the manuscript and prefer to submit it to another journal, notify me immediately so that the manuscript may be formally withdrawn from consideration by mSystems.

Revision Guidelines

Sincerely,
Anthony Fodor
Editor
mSystems

Reviewer #1 (Comments for the Author):

Critique of revised manuscript by Dalldorf et al for mSystems.

1. The authors have done a relatively good job at clarifying the main point of the study, i.e. that the RNAP substitutions in evolved strains that result in faster cell growth rates have transcriptomes characteristic of increased ribosomal expression ("greed" response) and a decreased RpoS stress response ("fear" response). This is of course expected, because faster growth rates require an increased rate of ribosome synthesis and that would cause a corresponding reduction in RpoS-dependent transcription. As I indicated in my initial review, the authors didn't do as good a job at addressing why the mutations cause the described changes in the transcriptome. Based on the positions of the substitutions in the structure of RNAP, I provided possible

explanations for how each of the substitutions might affect the conformation of the enzyme-DNA complex. In response, the authors have introduced a table in the supplemental data section that reproduces my suggestions almost verbatim, but they haven't taken that information and used it to construct a model for how the mutations might change the growth rate.

2. A straightforward model (perhaps too simplistic) is that the substitutions increase the stability of the promoter complex and/or the transcription elongation complex, either directly or indirectly. Since ppGpp binding to RNAP is thought to function by destabilizing rRNA-promoter complexes and elongation complexes, perhaps the substitutions immunize RNAP against the effects of ppGpp, either by directly interfering with ppGpp binding to the enzyme or by increasing the stabilities of RNAP-DNA complexes enough to bring them outside the range where ppGpp binding would reduce transcription. Perhaps acknowledgement of this kind of model in the Discussion might provide readers with an example of how the mutations could result in the increased growth rate phenotype.

3. Some minor issues:

The authors still focus on an analysis of how the substitutions might affect the stability of the enzyme itself or perhaps just the rpoB-rpoC interface. Please clarify. If the former, is the model that changing the stability of the enzyme could lead to higher RNAP content and faster growth? Since RNAP is a relatively stable molecule already, this seems unlikely. In any case it is not made clear how an altered rpoB-rpoC interface would result in an increase in the growth rate, or more specifically an increase in the fraction of RNAP devoted to ribosomal expression, accounting for the "greed" phenotype.

4. In response to reviewer 2s' comment 2, they focus on the expression of the r-proteins. rRNA and other non-coding RNAs were not covered by their expression analysis of course, but it should be noted that effects on expression of the r-proteins are more likely to be indirect effects of effects on rRNA expression that occur at the level of r-protein mRNA translation. Although ppGpp regulates r-protein transcription to some degree, most of the effects on r-protein expression are almost certainly indirect through effects on rRNA expression.

5. In the text corresponding to reviewer 1 comment 12 (on the GAD genes), they suggest that ppGpp binds tightly to RNAP. In reality, the ppGpp interaction with RNAP is rather weak and short-lived. The authors would be better served by just stating that ppGpp binds to RNAP (just delete "tightly").

6. In the text corresponding to response 2 of reviewer 2, the authors state "which genes RNAP translates". Shouldn't this be "which genes RNAP transcribes"?

7. In the response to reviewer 2 comment #4 they state again that the mutations affect RpoS and ribosomal genes through their effects on "protein interfaces". I think the model should be that the effects are ultimately through effects on RNAP-DNA interactions.

Reviewer #2 (Comments for the Author):

In this resubmission, Dalldorf et al., investigate the role of fear/greed tradeoff in RNAP mutations that were generated during adaptive laboratory evolution. They find that mutations result in large shifts in f/g tradeoffs in the RpoS and ribosomal genes, as well as condition-specific transcriptional adaptations. The analysis is quite complex as it involves analyzing transcriptome data across many different conditions and mutations.

The original paper had numerous issues in describing the work, clarity of writing, and overstating the findings. The reviewers pointed out numerous issues with the paper and the authors do a nice job of addressing some of the issues.

1) The authors indicate that the 4 iModulons that are critical in F/G tradeoffs, two primary and GadX and ppGpp, as they add a dimension to greed in introduction.

2) The authors describe why the other iModulons (Aenaero, translation, ppGpp) while positive have many genes of unknown function and may not be all clearly regular by growth or stress genes and that future analyses may include these.

3) Per comments from reviewer 1, the mutations listed are described in detail in Supp. Table 4. In addition the authors indicate that the modifications MAY change protein interfaces.

Remaining issue:

The authors add that the RNAP mutations might not have similar impact on tradeoffs in these species, which is true. However, the conclusions made should be limited. The cross-species comparison (Fig. 5) only finds F/G in 7 out of 12 bacteria (58%), and there are considerable differences in the gene composition of the fear group (Line 381-382). This hardly makes this a global property of bacterial transcription.

Lines - 38-39/387-388/444-445: should be limited to say : greed genes that are involved in f/g tradeoffs in e.coli can also be found in certain other species.

Reviewer #3 (Comments for the Author):

This revised manuscript does a much better job of explaining what the authors have done. This provides evidence that a variety of RNAP mutants, selected for improved growth under various conditions, lead to a shift of transcriptomes from "fear" RpoS regulons (dealing with various stresses) to "greed", increased translation patterns. While the basic observation was previously presented, this expands the previous analysis significantly. I had a few minor points that need attention.

1. The addition to Table S1 of a column labeled "Midpoint strains w/mutation/total strains mutation" is noted in the response to reviewers, but is not actually explained. A footnote or legend is needed to explain what this is.
2. Line 384: this says there is a "large explained variance". Should this be unexplained variance? If "explained" is correct, please indicate in what way this is explained.
3. Given that both ppGpp and GadX imodulons have correlations to RpoS, is it instructive that these two don't have any correlation to each other? Do different components of the RpoS imodulon lead to the RpoS correlations?

Reviewer #1

1. The authors have done a relatively good job at clarifying the main point of the study, i.e. that the RNAP substitutions in evolved strains that result in faster cell growth rates have transcriptomes characteristic of increased ribosomal expression ("greed" response) and a decreased RpoS stress response ("fear" response). This is of course expected, because faster growth rates require an increased rate of ribosome synthesis and that would cause a corresponding reduction in RpoS-dependent transcription. As I indicated in my initial review, the authors didn't do as good a job at addressing why the mutations cause the described changes in the transcriptome. Based on the positions of the substitutions in the structure of RNAP, I provided possible explanations for how each of the substitutions might affect the conformation of the enzyme-DNA complex. In response, the authors have introduced a table in the supplemental data section that reproduces my suggestions almost verbatim, but they haven't taken that information and used it to construct a model for how the mutations might change the growth rate.

We completely understand and agree with the importance of the fundamental question of the structural basis for the transcriptional responses we observe in our study.

We are hesitant to confidently propose a concrete structural model for all of our mutations as we do not feel we have sufficient evidence. We included the table of candidate mechanisms as a middle ground to postulate without overclaiming. One primary issue with proposing a model is that we did not observe a clear correlation between structural location of RNAP mutations and transcriptomic impact, so we can not confidently support a specific model for all mutations. However, we do now propose a model for our centrally located mutations in the discussion section, which is further discussed in the response to comment 3. This sentiment has been included in the discussion at lines 358-360:

“However, we have no clear mechanistic explanation as to why mutations distant from this central region, such as *rpoC* G1055V, have similar impacts to the transcriptome.”

We consider this work to be primarily a systems study, with structural analysis included to provide context for how certain mutations might elicit a transcriptional response. In that light, we chose to focus on the downstream systemic effect of how the transcriptional changes affect growth rate, as computed by the genome-scale model of metabolism and expression.

2. A straightforward model (perhaps too simplistic) is that the substitutions increase the stability of the promoter complex and/or the transcription elongation complex, either directly or indirectly. Since ppGpp binding to RNAP is thought to function by destabilizing

rRNA-promoter complexes and elongation complexes, perhaps the substitutions immunize RNAP against the effects of ppGpp, either by directly interfering with ppGpp binding to the enzyme or by increasing the stabilities of RNAP-DNA complexes enough to bring them outside the range where ppGpp binding would reduce transcription. Perhaps acknowledgement of this kind of model in the Discussion might provide readers with an example of how the mutations could result in the increased growth rate phenotype.

We agree with the reviewer that ppGpp binding is likely involved, based on PyRosetta stabilization predictions of the interface we proposed that the interface is destabilized by the mutations leading to transcriptional changes. Lines 163-165: “[The central cluster of mutations are] nearby to a ppGpp binding site which the mutations are also mostly predicted to destabilize (see Supplemental Table 1) and thus likely modify its regulatory role¹⁹ which is tightly connected to RpoS’s own activity³³.”

Providing additional alternatives would strengthen the paper by giving more context to the reader, and we will thus include a similar model to the reviewer’s suggestion as an alternative to the model we have proposed which is discussed in the response to comment 3. Ross, 2013 introduced RNAP mutations and measured ppGpp-RNAP complex concentrations. The mutations were found to decrease ppGpp’s ability to bind to RNAP and thus modify transcription.

The following has been added to the discussion at lines 363-364:

“Other RNAP mutations were found to eliminate the destabilizing effect of ppGpp binding to RNAP, thus reducing the inhibition of transcription by ppGpp⁵⁷.”

3. The authors still focus on an analysis of how the substitutions might affect the stability of the enzyme itself or perhaps just the rpoB-rpoC interface. Please clarify. If the former, is the model that changing the stability of the enzyme could lead to higher RNAP content and faster growth? Since RNAP is a relatively stable molecule already, this seems unlikely. In any case it is not made clear how an altered rpoB-rpoC interface would result in an increase in the growth rate, or more specifically an increase in the fraction of RNAP devoted to ribosomal expression, accounting for the "greed" phenotype.

The model our evidence best supports, at least for our centrally located mutations, is that a destabilized rpoB-rpoC interface creates a less stable open complex which affects transcription. Utrilla, 2016 found a destabilizing effect on this rpoB-rpoC interface and similar impacts to transcription as Conrad, 2010 who carried out a detailed kinetic study of a rpoC mutation that led to a destabilized open complex and increased transcription rate. The central group of our mutations occur near these previously studied mutations, have similar

impacts to the transcriptome, and are predicted to destabilize the rpoB-rpoC interface.

However, distant mutations such as rpoC G1055V have a highly similar impact to mutations near said catalytic core. Additionally, some mutations such as rpoB R200P have distinct transcriptional effects inferring a location-specific impact on the transcriptome for certain mutations. Detailed kinetic studies of these mutations would be necessary to put forth a comprehensive model so we do not have clear explanations for these outliers. We therefore want to provide proper context for our model and make it clear that other hypotheses exist.

This model is now directly outlined in the discussion section, while still providing additional alternative hypotheses. Lines 352-364 now read:

“A previous study compared two RNAP mutations², *rpoB* E672K and *rpoB* E546V, and found that they destabilize the rpoB-rpoC interface⁵⁴. Another study using *in vitro* assays linked an *rpoC* deletion from 3,611 to 3,619 bp (**near to the rpoB-rpoC interface**) to destabilizing the open complex of RNAP which led to decreased transcriptional pausing on the promoter, reduced RNAP’s open complex half-life, and increased elongation rates⁵⁵. **For our centrally located mutations, our evidence best supports this model of a destabilized rpoB-rpoC interface leading to a destabilized open complex thus causing transcriptional changes. However, we have no clear mechanistic explanation as to why mutations distant from this central region, such as rpoC G1055V, have similar impacts to the transcriptome.** The impact of RNAP mutations have also been shown to be similar to strains with reduced number of ribosomal operons, suggesting that these mutations are **possibly** modifying ribosomal availability and/or distribution⁵⁶. **Other RNAP mutations were found to eliminate the destabilizing effect of ppGpp binding to RNAP, thus reducing the inhibition of transcription by ppGpp⁵⁷.**”

4. In response to reviewer 2s' comment 2, they focus on the expression of the r-proteins. rRNA and other non-coding RNAs were not covered by their expression analysis of course, but it should be noted that effects on expression of the r-proteins are more likely to be indirect effects of effects on rRNA expression that occur at the level of r-protein mRNA translation. Although ppGpp regulates r-protein transcription to some degree, most of the effects on r-protein expression are almost certainly indirect through effects on rRNA expression.

We apologize, it is not clear what is being referred to here. Reviewer #2's second comment reads “As the authors important findings involve ppGpp and GaDX translation activity, they should describe the role of ppGpp and DkSA in stress regulation in the introduction (line 70). Move from line 252.” Said changes were made.

The only discussion of non-coding RNAs is in Reviewer #3's seventh comment, which talks about RNA-Seq library approaches and is primarily about computational details. This discussion, while interesting, is not included in the manuscript.

Non-coding RNAs can have large effects on transcriptional states and there is little doubt they play some role in the fear/greed tradeoff. As the reviewer states, said effect likely involves large complex regulatory networks occurring at multiple levels of the central dogma. We have not done analysis on the non-coding RNAs in this project, but they are likely of potential interest for future study and we implore others to utilize our public data to do so.

5. In the text corresponding to reviewer 1 comment 12 (on the GAD genes), they suggest that ppGpp binds tightly to RNAP. In reality, the ppGpp interaction with RNAP is rather weak and short-lived. The authors would be better served by just stating that ppGpp binds to RNAP (just delete "tightly").
"Tightly" has been removed from line 246.
6. In the text corresponding to response 2 of reviewer 2, the authors state "which genes RNAP translates". Shouldn't this be "which genes RNAP transcribes"?
"Translates" has been replaced with "transcribes" at line 78.
7. In the response to reviewer 2 comment #4 they state again that the mutations affect RpoS and ribosomal genes through their effects on "protein interfaces". I think the model should be that the effects are ultimately through effects on RNAP-DNA interactions.
Line 31-33 now reads: "We found that these single RNAP mutation strains resulted in large shifts in the f/g tradeoff primarily in the RpoS regulon and ribosomal genes, likely through modifying **RNAP-DNA interactions.**"

Reviewer #2

1. The authors add that the RNAP mutations might not have similar impact on tradeoffs in these species, which is true. However, the conclusions made should be limited. The cross-species comparison (Fig. 5) only finds F/G in 7 out of 12 bacteria (58%), and there are considerable differences in the gene composition of the fear group (Line 381-382). This hardly makes this a global property of bacterial transcription. Lines - 38-39/387-388/444-445: should be limited to say : greed genes that are involved in f/g tradeoffs in e.coli can also be found in certain other species.

We agree with the reader that 7/12 is probably not sufficient to call this a ubiquitous tradeoff yet (although we do note fear/greed tradeoffs may exist in the other organisms but have not yet been captured by our expression analysis, simply because not all expression compendia are as complete as that for E. coli).

Our claims regarding the generality of the tradeoff have been reduced as suggested, the following changes have been made:

Line 37-38 now reads: “A phylogenetic analysis found the **greed-related genes of the** tradeoff present in numerous bacterial species.”

Line 337-339 now reads: The presence of the **greed-related genes of the f/g** tradeoff across such a wide range of species implies that **they** may be a global property of bacterial transcriptomes.

Line 391-393 now reads: Finally, the phylogenetic distribution of the **greed-related genes of the f/g** tradeoff is broad, suggesting that this tradeoff may emerge as a universal feature of the bacterial transcriptome that can be captured by iModulons.

Reviewer #3

1. The addition to Table S1 of a column labeled "Midpoint strains w/mutation/total strains mutation" is noted in the response to reviewers, but is not actually explained. A footnote or legend is needed to explain what this is.

The legend for Supplemental Table 1 now reads:

“Supplemental Table 1: The mutations created for this study and the reason for their inclusion. **The column labeled “Midpoint Strains w/ Mutation / Total Strains w/ Mutation and Midpoints” refers to how often these particular mutations are found early in adaptive laboratory evolutions. The fractions listed are the number of corresponding midpoint strains that contain the mutation out of all independent lineages with said mutation. Only strains with both the mutation and midpoint samples available are considered.**”

2. Line 384: this says there is a "large explained variance". Should this be unexplained variance? If "explained" is correct, please indicate in what way this is explained.

We apologize for the lack of clarity in this line. Explained variance in this context refers to the amount of variance of the input expression data for ICA that an iModulon explains. An iModulon with higher explained variance typically is a larger iModulon of more importance and generally there is more confidence that it is a true signal and not experiment-specific changes and/or noise. More context for this can be found at iModulonDB.org.

Line 312-313 now reads: “Anaero-related, in addition to Translation and ppGpp, has a positive correlation with growth and a large explained variance of **PRECISE’s expression data.**”

3. Given that both ppGpp and GadX iModulons have correlations to RpoS, is it instructive that these two don't have any correlation to each other? Do different components of the RpoS iModulon lead to the RpoS correlations?

The genes of RpoS are all well correlated to both iModulons so it does not seem that ppGpp and GadX are correlating to different parts of RpoS. Several possible explanations exist for their lack of correlation. Most likely however, is that these correlations are due to iModulon calculations. P1K, an updated version of PRECISE 2.0 that was released after this manuscript was written, shows a stronger correlation between these two iModulons ($r^2 = 0.058$, $p_value = 1.7e-16$). These analyses and explanations, while interesting to us, are difficult to include in this manuscript without a lengthy discussion of ICA that would likely detract from the overall readability of this manuscript.

The two outlier genes in the upper left are *yjfl* and *yjfJ*, neither of which have functional annotations.

Re: mSystems00305-24R1 (The hallmarks of a tradeoff in transcriptomes that balances stress and growth functions)

Dear Prof. Bernhard O Palsson:

Thanks you for submitting the revised version and your careful attention to reviewer comments. Your manuscript has been accepted, and I am forwarding it to the ASM production staff for publication. Your paper will first be checked to make sure all elements meet the technical requirements. ASM staff will contact you if anything needs to be revised before copyediting and production can begin. Otherwise, you will be notified when your proofs are ready to be viewed.

Cover Image Submissions: If you would like to submit a potential Cover Image, please email a file and a short legend to msystems@asmusa.org. Please note that we can only consider images that (i) the authors created or own and (ii) have not been previously published. By submitting, you agree that the image can be used under the same terms as the published article. Image File requirements: TIF/EPS, 7.5 inches wide by 8.25 inches tall (at least 2,250 pixels wide by 2,475 pixels tall), minimum 300 dpi resolution (600 dpi preferred), RGB, and no figure elements, e.g., arrows or panel labels. The legend should be a short description of the image, 1-2 sentences recommended.

Congratulations on the acceptance of your manuscript and thank you for submitting your paper to mSystems.

Sincerely,
Anthony Fodor